# Inhibitory Receptors and Immune Checkpoints Regulating Natural Killer Cell Responses to Cancer

**DOI:** 10.3390/cancers13174263

**Published:** 2021-08-24

**Authors:** Irina Buckle, Camille Guillerey

**Affiliations:** Cancer Immunotherapies Laboratory, Mater Research Institute—The University of Queensland, Translational Research Institute, 37 Kent Street, Woolloongabba, QLD 4102, Australia

**Keywords:** NK cells, immune checkpoint, immunotherapy, cancer, exhaustion

## Abstract

**Simple Summary:**

Recent years marked the discovery and increased understanding of the role immune checkpoints play in immunity against cancer. This has revolutionized cancer treatment, saving the lives of many patients. For numerous years the spotlight of success has been directed towards T cells; however, it is now appreciated that other cells play vital roles in this protection. In this review we focused on cytotoxic lymphocytes Natural Killer (NK) cells, which are known to be well equipped in the fight against cancer. We explored the role of well-described and newly emerging inhibitory receptors, including immune checkpoints in regulating NK cell activity against cancer. The knowledge summarized in this review should guide the development of immunotherapies targeting inhibitory receptors with the aim of restoring NK cell responses in cancer patients.

**Abstract:**

The discovery of immune checkpoints provided a breakthrough for cancer therapy. Immune checkpoints are inhibitory receptors that are up-regulated on chronically stimulated lymphocytes and have been shown to hinder immune responses to cancer. Monoclonal antibodies against the checkpoint molecules PD-1 and CTLA-4 have shown early clinical success against melanoma and are now approved to treat various cancers. Since then, the list of potential candidates for immune checkpoint blockade has dramatically increased. The current paradigm stipulates that immune checkpoint blockade therapy unleashes pre-existing T cell responses. However, there is accumulating evidence that some of these immune checkpoint molecules are also expressed on Natural Killer (NK) cells. In this review, we summarize our latest knowledge about targetable NK cell inhibitory receptors. We discuss the HLA-binding receptors KIRS and NKG2A, receptors binding to nectin and nectin-like molecules including TIGIT, CD96, and CD112R, and immune checkpoints commonly associated with T cells such as PD-1, TIM-3, and LAG-3. We also discuss newly discovered pathways such as IL-1R8 and often overlooked receptors such as CD161 and Siglecs. We detail how these inhibitory receptors might regulate NK cell responses to cancer, and, where relevant, we discuss their implications for therapeutic intervention.

## 1. Introduction

Natural Killer (NK) cells are the cytotoxic members of the innate lymphoid cell (ILC) family [1]. They are well known for their ability to detect and kill virally infected, pre-malignant, and malignant cells [2]. Rapid serial killing of tumor cells by NK cells is dependent on lytic granules containing granzymes and perforin, while the Fas/FasL death receptor pathway contributes to late killing events [3]. Besides their tumor-killing activity, NK cells limit the dissemination and growth of metastases by sculpting tumor architecture [4] or maintaining tumor cells in dormancy [5]. Moreover, NK cells are important orchestrators of cancer immunity through the production of cytokines, chemokines, and growth factors that influence immune cells and stromal cells within the tumor microenvironment [6]. Of particular interest, two independent studies using mouse melanoma tumor models demonstrated NK cell ability to recruit and promote the differentiation and/or survival of type 1 conventional dendritic cells (cDC1), a subset of professional antigen-presenting cells specialized in CD8^+^ T cell priming [7,8]. Correlations analyses suggested that this NK cell/cDC1 axis might determine melanoma patient responsiveness to anti-PD1 immune checkpoint therapy [7].

A large number of preclinical studies support a protective role of NK cells in mouse cancer models where NK cells might be more efficient at limiting metastatic spread than controlling the growth of solid tumors [9]. In cancer patients, the historical association of high NK cell infiltration with positive prognostic might have been misled by the use of unspecific markers to identify NK cells [2]. Recent analysis of RNA transcript abundance of several NK cell-associated genes in 25 different cancer types revealed that, in cancers responsive to immune checkpoint blockade therapy, high levels of NK cell-related transcripts correlate with favorable prognostics [10]. By contrast, in some cancers, such as uveal melanoma or kidney renal clear cell carcinoma, this NK cell-related gene expression program is associated with deleterious patient outcomes [10]. In acute lymphoblastic leukemia, high percentages of cytokine-producing NK cells harboring an activated phenotype was shown to predict a poor clinical prognostic [11]. These new data highlight the importance of considering NK cell subsets and function in addition to absolute NK cell numbers and suggest that NK cells may behave differently across different cancer types. 

NK cells have emerged as attractive candidates for next-generation cancer therapies [12,13]. Unlike T cells, NK cells are not restricted by molecules of the major histocompatibility complex (MHC), a feature that is associated with a high safety profile of NK cellular therapies, even in the allogeneic setting. NK cells are also capable of eliminating tumor variants that may have escaped T cell control via the missing-self activation mechanism. Several NK cell-based therapies have made it to the clinic such as chimeric antigen receptor (CAR) NK cells, currently in clinical trials [13]. The generation of high numbers of efficacious CAR NK cells should be facilitated by recent progress in NK cell engineering [14] as well as the identification of targetable negative regulators of NK cell activity such as cytokine-inducible SH2-containing protein (CIS) [15,16].

Unfortunately, so far, NK cell-based therapies have been limited by their incapacity to overcome the immunosuppressive mechanisms at play in cancer patients [2]. A myriad of different pathways involving cancer cells, stromal cells, and immune-suppressive cells such as myeloid-derived suppressor cells (MDSCs) or regulatory T cells (Tregs) contribute to NK cell subset imbalance, alterations of NK cell phenotype, and dysregulation of NK cell functions in cancer. Transforming Growth Factor β (TGF-β) has been identified as a major negative regulator of NK cell activity [17], and, more recently, Activin-A was found to limit NK cell metabolic activity and decrease NK cell control of experimental metastasis in mice [18]. Further contributing to immunosuppression, immune checkpoints are key regulators of T cell responses that are now gaining increasing attention from the NK cell field [19].

In this review, we discuss regulation of NK cell functions in cancer through inhibitory receptors including immune checkpoints. We briefly introduce the role of classical human leukocyte antigen (HLA)-binding NK cell inhibitory receptors before focusing on receptors having recently attracted attention in the NK cell field. These include inhibitory receptors that recognize the molecules of the nectin and nectin-like family such as TIGIT or CD96, as well as immune checkpoints historically described on T cells such as PD-1 and TIM-3. We also discuss recently discovered pathways as well as receptors, which are often overlooked, such as Siglecs and Irp60.

## 2. Regulation of NK Cell Activation and Cytotoxic Activity

Human NK cells are conventionally defined as CD3^-^CD56^+^ lymphocytes, further divided into CD56^bright^ and CD56^dim^ NK cell subsets [20]. There is an increasing appreciation of the complexity of NK cell diversity that comprises a wide spectrum of phenotypes, maturation levels, and functional specializations [21]. Each NK cell subset expresses a selected set of cell-surface receptors that govern its ability to respond to diverse stimuli. Indeed, NK cell activation is determined through the integration of a variety of signals including cell-bound or soluble ligands, cytokines, and danger-associated molecular patterns (DAMPs) [2]. Activated NK cells produce cytokines such as IFN-γ, TNF-α, or GM-CSF and kill target cells such as cancer cells. NK cells distinguish their targets from healthy autologous cells through an array of germline-encoded cell surface receptors that provide either activating or inhibitory signals [22]. 

Additional regulatory mechanisms of NK cell functions were recently identified. For instance, Natural Killer Cell Granule Protein 7 (NKG7) controls cytotoxic granule exocytosis of NK cells [23]. Compared with WT mouse NK cells, Nkg7^−/−^ NK cells were shown to exhibit impaired degranulation against tumor cells, resulting in decreased in vivo protection against experimental B16F10 or LWT1 metastasis. New data also indicate that the spatial distribution of ligands for NK cell receptors influences the stability of NK cell contact with the potential target cell [24]. Within the immune synapse, NK cell degranulation was observed in areas of local signaling through activating NK cell receptors. For a detailed description of activating receptors and their ligands, the reader is invited to refer to previous reviews [25,26]. In this review, we will focus on inhibitory receptors (Figure 1).

## 3. Inhibitory Receptors Binding to MHC-I Molecules

NK cells express several inhibitory receptors characterized by the presence of immunoreceptor tyrosine-based inhibition motifs (ITIMs) in their cytoplasmic tail [26]. Tyrosine phosphorylation of ITIMs allows the recruitment of protein tyrosine phosphatases SHP-1 and SHP-2, leading to the transmission of inhibitory signals that limit the strength of signals received from activating receptors [27]. Classical NK cell inhibitory receptors bind to MHC molecules called HLA in humans [28]. Through these interactions, inhibitory NK cell receptors are responsible for the “missing-self recognition”, a process whereby NK cells recognize foreign or dangerous autologous cells that might have escaped adaptive immunity through down-regulation of surface class I MHC (MHC-I) molecule expression [29]. 

MHC-I-specific NK cell receptors have been extensively reviewed elsewhere [28] and will be only briefly discussed here. The heteromeric receptor CD94/NKG2A that binds to the conserved non-classical MHC molecule HLA-E is the earliest inhibitory receptor expressed by developing and immature NK cells [30]. As they mature, NK cells upregulate their expression of Killer Ig Inhibitory receptors (KIRs), a group of highly polymorphic receptors transmitting activating or inhibitory signals [28]. Inhibitory KIRs are characterized by 2 or 3 Ig-like extracellular domains and long cytoplasmic tail; they recognize allotypic determinants shared by diverse groups of class I HLA molecules. NK cells also express Ig-like transcript 2 (ILT2, LILRB1, or L1R1), which has a broad specificity for HLA-I molecules [31]. Most individuals have a unique KIR repertoire [32]. Although KIRs and HLA genes are co-inherited, they independently segregate, leading to variable KIR-HLA interactions between individuals. Moreover, within a given individual, each single NK cell expresses its own repertoire of NKG2A, ILT2, and KIRs inhibitory receptors. The self-reactivity of heterogeneous NK cell populations and their ability to perform missing-self recognition of aberrant cells is controlled by a process called NK cell education. As a result of NK cell education, NK cells expressing inhibitory receptors specific for self-MHC-I are “licensed” and more reactive than “unlicensed” NK cells when triggered through activating receptors [33].

Strategies have been implemented to block KIRs and NKG2A in cancer patients to trigger missing-self recognition of malignant cells [13,34]. So far, KIR antagonists such as IPH2101 or IPH21022 (Lirilumab, Bristol-Myers Squibb) that block the interaction between KIR2DL-1,-2,-3 inhibitory receptors and their ligands have generated disappointing results [13]. Lack of efficacy has been explained by monocyte-mediated trogocytosis of KIR2D molecules from NK cell surface leading to NK cell hypo-responsiveness [35]. More encouraging results have been obtained with monalizumab (IPH2201, InnatePharma/AstraZeneca), a humanized anti-NKG2A IgG4 monoclonal antibody (mAb) that has been safely administered to cancer patients [30]. The very little clinical activity of monalizumab monotherapy in gynecologic cancers [36] aligns with data from preclinical models where anti-NKG2A mAbs alone showed no efficacy against subcutaneous mouse tumors [37,38]. However, in these models, NKG2A blockade was shown to enhance the protective effect of peptide vaccination or anti-PD-L1 mAbs [37,38]. Confirming the potential of combining monalizumab with other therapeutics, the combination of monalizumab with cetuximab (an anti-EGFR blocking mAb) showed a 31% objective response rate in patients with squamous cell carcinoma of the head and neck, an outcome that compares favorably against historical data from patients receiving cetuximab alone [38]. Currently, eight clinical trials are investigating monalizumab as a single agent or in combination with anti-EGFR or anti-PD-L1 mAbs, tyrosine kinase inhibitors, and chemotherapy in a range of solid and hematological malignancies [6].

KIRs and NKG2A have been referred to as “NK cell immune checkpoints” for their ability to inhibit NK cell functions and their potential as therapeutic targets, in analogy to the T cell-associated immune checkpoints PD-1 and CTLA-4 [34,35]. In light of the revolutionizing outcomes obtained with the blockade of PD-1 or CTLA-4 pathways, the limited efficacy of KIR-blockade in cancer patients is underwhelming. However, the biological functions of KIRs on NK cells differ from those of T cell-associated immune checkpoints. T cell-associated immune checkpoints are up-regulated on T cells upon activation. They are associated with chronic stimulation and exhaustion and contribute to limit the intensity of an ongoing immune response. By contrast, KIRs and NKG2A are expressed on NK cells at steady state, mediate tolerance to self in the absence of pathology, and contribute to NK cell education. As such, disturbance of NK cell education by blocking the dynamic interactions of KIR2D with their HLA ligands has been proposed as a mechanism explaining the poor efficacy of IPH2101 [35]. While monalizumab yielded positive outcomes, the anti-tumor effect of NKG2A blockade in pre-clinical models has been attributed to both T cells and NK cells [37,38]. Interestingly, NKG2A expression profile differs between CD8^+^ T cells and NK cells; while NKG2A is expressed on approximately 50% of blood-circulating NK cells, it is virtually absent in peripheral CD8^+^ T cells in healthy individuals but is up-regulated during chronic viral infections as well as in tumor-infiltrating T cells [30,34]. Therefore, NKG2A expression profile on T cells resembles those of other T cell-associated immune checkpoints. The relative importance of NK cell and T cell contributions to the clinical efficacy of monalizumab remains to be evaluated. 

## 4. Inhibitory Receptors Binding to Nectin and Nectin-Like Molecules

Recently, a group of receptors that regulate NK cell functions through interacting with molecules of the nectin and nectin-like family have entered the spotlight and have been referred to as novel NK cell immune checkpoints [39]. This group of receptors includes DNAM-1, known to mediate NK cell activation [9], TIGIT, and CD112R, which are two inhibitory receptors [40,41,42], and CD96, the exact function of which remains controversial [43,44]. These receptors bind to CD155 (Nexl-5 or PVR) and CD112 (Nectin-2), two ligands ubiquitously expressed at low levels and often up-regulated on cancerous cells [45,46]. Moreover, CD155 and CD112 are expressed on myeloid cells within the tumor microenvironment [47,48].

### 4.1. TIGIT

TIGIT (also called VSig9, Vstm3, or WUCAM) is expressed on 40–60% of human peripheral blood NK cells, with higher expression levels reported on the CD56^dim^ NK cell population compared to CD56^bright^ NK cells [49,50,51]. This suggests that TIGIT expression correlates with maturation, a pattern that is reminiscent of mouse data indicating progressive acquisition of TIGIT with ontogeny [52]. Up-regulation of TIGIT expression following activation has been observed on both mouse and human NK cells [44,53]. However, conflicting data have been reported in the context of chronic inflammation. Increased TIGIT expression has been observed on peripheral blood NK cells of patients with HIV infection [49] or gastrointestinal cancer [54] compared with healthy donors. By contrast, TIGIT levels are decreased on blood NK cells from patients with autoimmune diseases [50] or from lymphoma patients [51]. Decreased levels of TIGIT were also observed on melanoma- or hepatocellular carcinoma-infiltrating NK cells when compared with circulating or peritumoral NK cells [55,56]. Overall, TIGIT pattern of expression differs from conventional immune checkpoints such as PD1 or CTLA-4 that are up-regulated with chronic stimulation. This is further supported by the observation that adaptive NK cells present in CMV-infected individuals show decreased TIGIT expression compared to conventional NK cells [48]. 

Importantly, TIGIT expression on NK cells does not necessarily correlate with exhaustion. Although mouse TIGIT^+^ NK cells infiltrating CT26 tumors appear less functional than their TIGIT^-^ counterparts, as shown by decreased production of IFN-γ and TNF-α as well as decreased degranulation [54], it should be noted that CT26 tumor cells express TIGIT ligand CD155 [57]. In this regard, while mouse TIGIT^+^ NK cells are less responsive to CD155^+^ tumors, they respond better than TIGIT^-^ NK cells to CD155-unrelated stimuli [52]. These data indicate a role for TIGIT in mouse NK cell education. In melanoma patients, circulating TIGIT^+^ NK cells exhibit lower killing activity against the CD155^+^ melanoma cell line FO-I despite harboring higher expression of granzymes A and B and perforin compared with TIGIT^-^ NK cells [55]. Overall, data in mice and humans indicate that TIGIT^+^ NK cells are functional and TIGIT acts as an inhibitory receptor that limits NK cell responses to CD155^+^ target cells. 

TIGIT might inhibit NK cell functions through diverse mechanisms, reviewed in [40]. TIGIT mediates direct inhibitory signaling in NK cells through ITIM and immunoreceptor tail tyrosine (ITT)-like motifs present in its cytoplasmic tail, as shown with the YTS human NK cell line [58,59,60]. TIGIT may also compete with the activating receptor DNAM-1 for ligand binding, although formal evidence for this mechanism being at play in NK cells is still missing [61]. Still, recent data from Chauvin et al. support a competition mechanism in human NK cells. They observed that membrane-bound CD155 stimulation induced down-regulation of both TIGIT and DNAM-1 expression. This phenotype was associated with NK cell dysfunction, thereby indicating that release of TIGIT-mediated inhibition is ineffective in the absence of DNAM-1 signaling [55]. Interestingly, both TIGIT and DNAM-1 expression could be rescued by IL-15 treatment, and combination with TIGIT blockade further enhanced human NK cell killing of the melanoma cell line FO-1 [55]. Finally, TIGIT binding to CD155 may trigger CD155 intracellular signaling in DCs, leading to the secretion on tolerogenic cytokines [61]. However, so far, there is no indication of NK cell contribution to this mechanism.

A large number of data supports the hypothesis that blocking TIGIT would promote NK cell functions against cancer. TIGIT blockade was shown to enhance human NK cell production of IFN-γ [49] as well as tumor cell killing, especially in the context of CD155 expression [53,55]. TIGIT blockade also prevents human NK cells from being inhibited by CD155-expressing myeloid-derived suppressor cells (MDSCs) [48]. Zhang et al. used Tigit^fl/fl^Ncr1^iCre/+^ mice to show that NK cell-targeted TIGIT deletion led to decreased cancer control in vivo [54]; however, these results contrast with those from another group who observed no difference between TIGIT^−/−^ and WT mice in the same model of B16F10 experimental metastasis [44]. Furthermore, Zhang et al. showed that anti-TIGIT-blocking mAbs induce NK cell-dependent protection against CT26 or B16F10 subcutaneous tumors as well as experimental metastasis [54]. In the same line, when combined with anti-IL-15/IL-15Rα complexes to expand NK cells, anti-TIGIT mAbs were found to induce protection against established B16F10 or LWT1 lung metastases in mice [55]. In summary, TIGIT is an inhibitory receptor that is highly expressed on human NK cells in both health and disease, and, in the presence of CD155-expressing cells, blocking TIGIT appears as a promising strategy to increase NK cell activity. 

### 4.2. CD96

CD96, also named T cell-activated increased late expression (TACTILE), is expressed on both mouse and human NK cells at steady state [44,62], with higher expression levels on human CD56^bright^ compared to CD56^dim^ NK cells [51,53]. Increased CD96 expression on human NK cells has been reported following treatment with IL-15 or TGF-β [53,56,63]. By contrast, down-regulation of CD96 on mouse NK cells was described at early time points of an LPS challenge inducing acute inflammation [44]. In cancer patients, increased CD96 expression has been observed on NK cells from ovarian cancer ascites compared to peripheral blood NK cells from healthy donors [53]. Furthermore, in the liver of hepatocellular carcinoma patients, NK cells from intra-tumoral regions express higher level of CD96 compared to NK cells from peritumoral regions [56]. However, decreased percentages of CD96^+^ NK cells have been reported in the peripheral blood of pancreatic cancer patients when compared with healthy donors [64]. Interestingly, a recent publication suggested that platelet-cloaked tumor cells might suppress expression of CD96 on the NK cell surface [63]. Overall, reports agree on CD96 being expressed on NK cells in the absence of disease; yet, it is unclear how activating stimuli or chronic inflammation such as occurring in cancer might alter CD96 expression levels. 

CD96 binds to CD155 with an affinity lower than TIGIT but higher than DNAM-1 [65]. To date, the role of CD96 in lymphocytes is not clearly established, as both activating and inhibitory functions have been reported. It should be noted that, while both mouse and human CD96 harbor an ITIM domain in their cytoplasmic tail, only the human cytoplasmic tail contains a YXXM motif [66]. The presence of an YXXM may confer either inhibitory or activating potential for human CD96 [67]. CD96 was initially described as a cell-adhesion molecule that was shown to enhance the cytotoxicity of freshly established human NK cell lines in redirected lysis assays against P815 cells [68]. These results were confirmed by others who observed that CD96 slightly increased the cytotoxicity of IL-2-activated human NK cells when co-engaged with 2B4 or NKp30 [58]. However, blocking CD96 through mAbs had no effect on human NK cell degranulation or IFN-γ production against various cancer cell lines [53,62,63]. Similarly, no difference in killing has been observed between CD96^+^ and CD96^-^ NK cells from healthy donors [56]. Finally, mouse CD96^−/−^ NK cells were found as competent as their WT counterpart in in vitro killing assays [44]. It is possible that a role for CD96 might only be revealed in the presence of CD155. In vitro, NK cells from CD96^−/−^ mice produce more IFN-γ than WT NK cells only if cultured in CD155-Fc-coated plates or in the presence of CD155-expressing DCs [44]. A significant body of work from the Smyth group performed in various mouse cancer models suggests that CD96 might limit NK cell-mediated protection against metastases and, therefore, blocking CD96 would enhance NK cell production of IFN-γ and cancer control [44,69,70]. While these data are in line with a study showing anti-tumor activity of anti-CD96 mAbs against a human clear cell carcinoma cell line xenotransplanted in SCID beige mice [71], they are opposed by a recent report indicating that CD96 crosslinking activates mouse and human CD8^+^ T cells [43]. This latest evidence would advocate against the use of CD96 blocking in cancer patients. In hepatocellular carcinoma, patients with higher percentages of intra-tumoral CD96^+^ NK cells show reduced disease-free survival [56]. A better understanding of CD96 prognostic value in different cancer types as well as a better knowledge of CD96 signaling pathways in both NK cells and CD8^+^ T cells should help reconcile so far contradicting data and establish strategies to target this receptor in cancer patients.

### 4.3. CD112R

CD112R (or PVRIG) is a newly identified lymphocyte receptor that binds to CD112 with much higher affinity than DNAM-1 or TIGIT [72]. In the absence of pathology, CD112R is expressed on 5–15% of mouse NK cells [73] and on both CD16^+^ and CD16^-^ human NK cell subsets [41,72]. Moreover, CD112R was found to be upregulated on mouse NK cells infiltrating subcutaneous tumors [73]. In this study, CD112R expression on mouse NK cells correlated with higher expression of the inhibitory receptors CD96, TIGIT, TIM-3, and PD-1. In prostate cancer patients, CD112R is highly expressed on tumor-infiltrating NK cells [47], and, in endometrial cancer, CD112R and TIGIT are co-expressed on NK cells [47]. The addition of neutralizing anti-CD112R antibodies to human NK cells co-cultured with MDA, a human breast cancer cell line expressing CD112 and CD155, increased NK cell degranulation and production of IFN-γ [41]. These data established CD112R as an inhibitory receptor for NK cells, a finding that was recently confirmed by in vivo experiments [73]. Li et al. showed that CD112R deficiency or treatment with blocking mAbs against CD112R protected mice against subcutaneous MC38 tumors, and this was associated with increased degranulation and IFN-γ production by tumor-infiltrating NK cells [73]. In human T cells, TIGIT and CD112R have been suggested to mediate two non-redundant inhibitory pathways, with TIGIT predominantly binding to CD155 and CD112R binding to CD112 [47]. It is tempting to speculate that a similar mechanism is at play in NK cells. Supporting this hypothesis, when compared with neutralization of TIGIT or CD112R individually, dual blockade of these receptors resulted in improved human NK cell antibody-dependent cellular cytotoxicity (ADCC) against trastuzumab-coated breast cancer cells in vitro [41]. 

## 5. T Cell-Associated Immune Checkpoints

Immune checkpoints were initially described as inhibitory receptors up-regulated on chronically activated T cells to allow return to homeostasis and prevent over-inflammation and tissue damage [74]. This class of receptors can be targeted through mAbs to restore immune responses in cancer patients, a therapeutic strategy that has led to unprecedented long-term responses in a variety of cancers [75]. In T cell, expression of immune checkpoints such as PD-1, CTLA-4, TIM-3, TIGIT, or LAG3 is associated with functional exhaustion [76]. Although some of these receptors have been observed on NK cells, their implications for NK cell functions are only starting to be explored. 

### 5.1. Members of the B7-CD28 Superfamily

#### 5.1.1. PD-1

The programmed cell death protein 1 (PD-1; CD279) is an immune checkpoint expressed on T cells, B cells, NK cells, and some myeloid cells [77]. PD-1 ligands, PD-L1 and PD-L2, are upregulated in a wide range of malignancies and their engagement results in down-regulation of T cell responses and immune escape [78]. The PD-1/PD-L1 axis blockade revolutionized cancer treatment and somewhat monopolized the cancer immunotherapy field. Despite the spotlight being primarily focused on T cells, recent data indicate a possible contribution of PD-1 expressing NK cells to this therapeutic success.

While naïve mouse NK cells do not express PD-1 [79,80,81], a fraction of PD-1^+^ tumor-infiltrating NK cells has been observed in various mouse subcutaneous and spontaneous genetically induced mouse cancer models [79]. Mouse NK cells have also been shown to up-regulate PD-1 following murine cytomegalovirus (CMV) infection [81]. These data have been contested by Judge et al., who reported no PD-1 expression on in vitro activated or tumor-infiltrating mouse NK cells and claimed that staining artifacts may have been caused by cell preparation or a low number of events [80]. However, it should be noted that induction of PD-1 expression on mouse NK cells might only be obtained under specific stimulatory conditions. For instance, IL-15 and IL-18 together with corticosterone, but not IL-12, were found to up-regulate PD-1 on mouse NK cells in vitro [81]. Moreover, the level of PD-1 up-regulation on mouse NK cells varies across tumor models, with approximately 40% of PD-1^+^ NK cells detected in the tumor infiltrate of NK cell-sensitive tumors RMA-S but very low PD-1 expression on B16 melanoma-infiltrating NK cells [79].

In healthy individuals, PD-1 expression is restricted to the terminally differentiated CD56^dim^NKG2A^-^KIR^+^CD57^+^ NK cell subset [82]. Despite PD-1^+^ NK cells being only observed in subjects serologically positive for human CMV, there is no correlation of expression between PD-1 and NKG2C [82], indicating that PD-1 is not specifically expressed on CMV memory NK cells. Up-regulation of PD-1 expression on NK cells has been reported in patients with ovarian carcinoma [82], Kaposi sarcoma [83], renal cell carcinoma [84], Hodgkin lymphoma [85], and multiple myeloma [86] as well as a number of digestive cancers [87]. Interestingly, while PD-1 expression was confined to CD56^dim^ NK cells in renal cell carcinoma patients [84] and Kaposi sarcoma patients [83], both CD56^bright^ and CD56^dim^ NK cells were found to up-regulate PD-1 in patients with hepatocellular carcinoma, esophageal squamous cell carcinoma, colorectal cancer, and other gastro-intestinal cancers [87]. Reported PD-1 up-regulation on activated human NK cells has been recently disputed by Judge et al., who used qRT-PCR and flow cytometry to show very limited PD-1 expression on IL-2- or IL-15-stimulated healthy donor NK cells [80]. They also failed to detect PD-1 expression on NK cells from round cell carcinoma and colon cancer patients [80]. However, it should be noted that the cancer patient cohort was small in this study (n = 7). Moreover, similarly to mouse NK cells, very specific stimuli seem to be required for induction of PD-1 expression on human NK cell surface, as Quatrini et al. demonstrated that only the combination of dexamethasone together the IL-12-, IL-15-, and IL-18-stimulating cytokines induced PD-1 up-regulation on healthy donor NK cells [88]. In addition to confirming a role for glucocorticoids in controlling PD-1 expression in both mouse and humans NK cells, this study highlights the potential effect of corticosteroid treatment on PD-1 expression levels in cancer patients NK cells. Although detection of PD-1-encoding mRNA in human NK cells indicates endogenous production [88,89], an interesting non-peer-reviewed preprint suggests that NK cells might also acquire PD-1 expression exogenously through membrane exchange with PD-1-expressing tumor cells [90].

PD-1 is a marker commonly associated with T cell exhaustion [91]. Yet, for mouse NK cells this does not seem to be the case. Indeed, when compared with their PD-1^-^ counterparts, PD1^+^ NK cells infiltrating mouse subcutaneous tumors showed increased responsiveness to receptor triggering ex vivo [79]. Considering that PD-1 up-regulation was associated with Sca-1 and CD69 expression, this study indicates that, in mice, PD-1 is expressed on activated and highly functional NK cells. In humans, data obtained using NK cells isolated from Kaposi sarcoma patients indicate that PD-1^+^ NK cells exhibit functional features of exhaustion, as shown by decreased degranulation and IFN-γ production compared to their PD-1^-^ counterparts [83]. However, other studies have attributed these observations to differential receptor expression between PD-1^+^ and PD-1^-^ NK cells subsets [82]. As such, decreased responsiveness to K562 target cells or receptor cross-linking might be caused by low expression of the activating receptors NKp46 and NKp30 on the CD56^dim^NKG2A^-^KIR^+^CD57^+^ PD-1^+^ NK cell subset [82,92]. Moreover, the low proliferation of healthy donor PD-1^+^ NK cells in response to IL-2 or IL-15 can be explained by the terminally differentiated profile of this specific subset [82]. Further arguing against an exhausted state of PD-1^+^ NK cells, when stimulated with PMA and ionomycin, PD-1^+^ NK cells from esophageal squamous cell carcinoma patients showed increased degranulation compared to their PD-1- counterparts [87]. 

A convincing body of evidence indicates that PD-1 might inhibit mouse and human NK cell functions. In vitro, PD-1 cross-linking was shown to limit the production of IFN-γ and TNF-α by NK cells isolated from pleural tumors [93]. Moreover, enhanced in vitro degranulation of PD1^+^ NK cells from ovarian patients against the OVCAR-5 carcinoma line was observed upon addition of anti-PD-L1/2 [82]. In vitro, PD-1 blockade was shown to improve the ability of NK cells from Hodgkin lymphoma patients to mediate direct killing and ADCC against the PD-L1-expressing cell line HDLM2 [85]. In pediatric post-transplantation lymphoproliferative disorders and in multiple myeloma, in vitro experiments established that PD-1 blockade improves NK cell response to autologous tumor cells [86,94]. Finally, in vivo experiments using mouse cancer models have highlighted the potential therapeutic benefit of blocking PD-1 on NK cells. Hsu et al. demonstrated that NK cells were essential for the therapeutic efficacy of PD-1/PD-L1 pathway blockade against RMA-S or CT26 mouse tumors engineered to express PD-L1 [79]; and Huang et al. used a mouse glioma model to illustrate the ability of anti-PD-1 mAbs to improve the efficacy of adoptive NK cell transfer therapy [95]. 

#### 5.1.2. CTLA-4

Cytotoxic T lymphocyte-associated antigen 4 (CTLA-4) is an immune checkpoint expressed on lymphocytes that has mostly been studied on Tregs and activated effector T cells. CTLA-4 limits T cell activation by competing with the costimulatory receptor CD28 for its ligands CD80 (B7-1) and CD86 (B7-2) on antigen-presenting cells [96]. 

Mouse NK cells do not express CTLA-4 at steady state but they were found to up-regulate CTLA-4 following prolonged stimulation to IL-2, and CTLA-4^+^ NK cells were also observed in mouse subcutaneous tumors [97]. To investigate a direct role of CTLA-4 in regulating NK cell functions, mouse NK cells were stimulated in vitro with plate-bound B7-1-IgG fusion protein [97]. CTLA-4^+^ NK cells isolated from WT mice showed impaired IFN-γ production compared to their CTLA-4^-^ counterparts, and the IFN-γ response of NK cells from CTLA-4^−/−^ mice was improved compared to those from WT mice. Interestingly, in these experiments, NK cells from CD28^−/−^ mice failed to produce IFN-γ, indicating that, similarly to what has been reported for effector T cells, CTLA-4 regulates NK cell responses through the inhibition of CD28-mediated stimulatory signal [97].

Contrasting with data on mouse NK cells, strong evidence for a direct role of CTLA-4 in regulating human NK cells is still lacking. Lougaris et al. observed CTLA-4 staining by flow cytometry on healthy donor NK cells stimulated with IL-2, with or without IL-12 and IL-18 [98], but others failed to detect any expression in similar conditions [19,99]. A recent non-peer-reviewed preprint claimed that, despite the absence of observable staining by flow cytometry, human NK cells expressed CTLA-4, as suggested by Western blot and qRT-PCR analysis as well as Fc-receptor-independent binding of anti-CTLA-4 mAbs to the CD16-negative NK92 cell line [100]. However, this report did not analyze surface expression of CTLA-4 on non-transformed human NK cells. As such, there is still no consensus on whether human NK cells might express CTLA-4 under specific activating conditions.

Nevertheless, a convincing body of work suggests that NK cells might be involved in the therapeutic success of anti-CTLA-4 mAbs, although probably through indirect mechanisms. A study utilizing the B16-OVA mouse melanoma model demonstrated that the anti-tumor effect of an IL-2/anti-IL-2 complex combined with anti-CTLA-4 mAbs was dependent on both NK cells and CD8^+^ T cells [101]. By contrast, NK cells were not required for treatment efficacy of IL-2/anti-IL-2 complex combined with anti-PD-1 mAbs, which relied uniquely on CD8^+^ T cells. The authors suggested that Treg depletion triggered by anti-CTLA-4 mAbs might be responsible for the enhancement of NK cell activity and therapeutic efficacy in this model. Data support similar mechanisms of action in cancer patients treated with the anti-fully human IgG1 anti-CTLA-4 mAb Ipilimumab. Indeed, an abstract submitted at the 2015 ASCO meeting reported that improved ex vivo NK cell cytotoxicity of melanoma patients treated with ipilimumab was associated with positive clinical responses [102]. In the same line, compared with short-term survivors, long-term survivor melanoma patients treated with Ipilimumab were shown to display enhanced degranulation in response to K562 target cells [103]. Due to its ability to induce monocyte-mediated ADCC against CTLA-4^+^ Tregs [104], ipilimumab might release Treg competition for the NK cell-activating cytokine IL-2, thereby indirectly improving NK cell functions [105,106,107,108]. Interestingly, restoration of CD56^dim^/CD56^bright^ NK cell ratio to physiological levels was observed in malignant mesothelioma patients treated with Tremelimumab, a fully human IgG2 anti-CTLA-4 mAb that does not induce Treg depletion [109,110]. Taken all together, scientific results so far suggest that, although human NK cells might present no or very low expression of CTLA-4, anti-CTLA-4 mAbs might indirectly promote NK cell activity through their induction of cytokine secretion by effector T cells and/or deletion of Tregs.

#### 5.1.3. BTLA (CD272)

B and T cell lymphocyte attenuator (BTLA/CD272) is a CD28/B7 superfamily member structurally and functionally similar to CTLA-4 and PD-1 [111]. BTLA binds to Herpes virus entry mediator (HVEM), a ligand expressed on various immune subsets [112]. HVEM is also recognized by the activating receptor CD160, expressed on human CD56^dim^ NK cells [113]. BTLA is expressed at very low levels on peripheral blood NK cells from healthy donors [113], but is up-regulated on NK cell surface in chronic lymphocytic leukemia (CLL) patients [114]. Interestingly, high BTLA expression on NK cells has been associated with poor outcome for CLL patients [114]. Incubation of peripheral blood mononuclear cell cultures from CLL patients with blocking anti-BTLA mAbs was found to increase NK cell IFN-γ production in response to PMA and ionomycin stimulation and cytotoxicity against the B-CLL cell line MEC-1 [114]. However, these experiments did not establish whether improved NK cell functions were caused by inhibition of BTLA signaling in NK cells or whether other cell types present in the culture may have confounded the results. Using NK92 cells engineered to express BTLA, Šedý et al. showed that BTLA signaling impairs NK92 cell killing of HVEM-expressing target cells [114]. Future experiments should investigate whether BTLA signaling directly inhibits primary human NK cells. It also remains to be determined whether BTLA up-regulation on NK cells is a specific feature of CLL or whether this observation might be generalized to other hematological cancers and, potentially, solid cancers. 

#### 5.1.4. Receptor(s) to B7-H3

B7-H3 (CD276) is a recently discovered member of the B7-CD28 superfamily [115] and is described as an orphan ligand [116] with several receptor candidates currently under investigation [117,118,119,120]. B7-H3 expression was reported on tumor cell lines, tumor-infiltrating DCs, and macrophages [121,122]. An early study described B7-H3 expression on human CD56^+^ NK cells stimulated with PMA and ionomycin [123]; however, no further reports have confirmed B7-H3 expression on either mouse or human NK cells. 

Recently, Lee et al. showed that NK cells, together with CD8^+^ T cells, contribute to the protection against E.G7 subcutaneous tumors or B16 experimental metastasis observed in B7-H3-deficient mice or in WT mice treated with antagonistic anti-B7-H3 mAbs [124]. Moreover, they analyzed NK cell cytotoxic activity following co-culture with DCs isolated from WT or B7-H3-deficient mice and showed that B7-H3-deficient DCs induced more potent NK cell-mediated killing. Expending these findings to humans, the authors found that B7-H3-Ig reduced degranulation of healthy donor NK cells in response to CD16 or NKG2D cross-linking. Overall, data indicate that B7-H3 expression on antigen-presenting cells or tumor cells might inhibit both mouse and human NK cells through binding to a yet unknown NK cell receptor. 

#### 5.1.5. VISTA

V-domain Ig suppressor of T cell activation (VISTA) is a potent negative regulator of T cell function [125]. Negligible VISTA expression has been reported on murine bone marrow and splenic NK cells [126,127], and low VISTA expression levels have been observed on healthy human CD56^dim^ NK cells [125]. VISTA is, therefore, unlikely to play a major role in the regulation of NK cell responses. 

### 5.2. LAG-3

Lymphocyte Activation Gene-3 (LAG-3) was originally reported to be selectively transcribed in activated T cells and NK cells [128]. Published data to date indicate minimal LAG-3 protein expression on the surface of unstimulated mouse and human NK cells [129,130,131]. In an early study, Miyazaki et al. used an antiserum to LAG3 and reported LAG-3 expression on polyI:C-stimulated mouse NK cells [132]. Since then, LAG-3 expression on mouse NK cells has been observed in mice deficient for the Wiskott–Aldrich syndrome protein [131]. Using peripheral blood NK cells from healthy HCMV-seropositive donors, Merino et al. showed that LAG-3 expression could be induced on human NK cells by a 7-day in vitro stimulation with IL-15 and NKG2C ligation [129]. Similarly, a non-peer-reviewed preprint indicates that LAG-3 is up-regulated on healthy donor NK cells stimulated for 7 days with IL-12 and IL-18 [133]. In these two studies, highest levels of LAG-3 expression were detected in the NKG2C^+^ NK cell subset, with up to 80% of chronically stimulated NKG2C^+^ NK cells expressing LAG-3. Another non-peer-reviewed preprint indicates that LAG-3 could also be induced on healthy donor NK cells upon 2-day culture with IL-2 and IFNα [130]. In agreement with the aforementioned studies, this last report suggests that LAG-3 is preferentially expressed on CD56^dim^CD16^+^KIR^+^CD57^+^NKG2C^+^ NK cells, representing a terminally differentiated subset and/or memory NK cells. 

Data from Merino et al. established that LAG-3 expression might be induced on chronically stimulated NK cells undergoing epigenetic reprogramming associated with exhaustion [129]. Supporting this idea, the authors showed that PD-1, which is commonly associated with T cell exhaustion, is expressed by a subset of LAG-3^+^ NK cells but not on their LAG-3^-^ counterparts. Moreover, when stimulated with K562 target cells, LAG-3^+^ NK cells produced reduced levels of IFN-γ [129]. Nevertheless, LAG-3^+^ NK cells retained their degranulation ability, indicating that their cytotoxic function is preserved [129]. In addition, seahorse assay analyses revealed increased glycolytic activity in LAG-3^+^ NK, suggesting a metabolically mature but not exhausted phenotype [130]. 

About 15 years ago, Miyazaki et al. used Lag-3-deficient mice to provide the first evidence for a role for LAG-3 in regulating NK cell functions. Surprisingly, they found that Lag-3 null mutation resulted in decreased NK cell cytolytic activity against Yac-1, IC-21, and J77 tumor cell lines, suggesting that LAG-3 might act as a co-stimulatory receptor on mouse NK cells [132]. So far, no other group has confirmed this activating role for LAG-3 in mice; however, it is clear that LAG-3 has no such role in human NK cells [134]. Overall, LAG-3 does not seem to regulate human NK cell cytotoxicity [130,134]. However, non-peer-reviewed findings from Narayanan et al. indicated that LAG-3 blockade enhances human NK cell production of IFN-γ, TNF-α, MIP-1α, and MIP-1β in response to THP1 target cells in vitro [130].

Taken as a whole, data to date indicate that LAG-3 might be up-regulated on chronically stimulated human NK cells and might negatively regulate NK cell production of pro-inflammatory cytokines.

### 5.3. TIM-3

T cell immunoglobulin and mucin domain 3 (TIM-3) is a co-inhibitory receptor first described two decades ago as a surface molecule expressed by IFN-γ-producing CD4^+^ and CD8^+^ T cells [135]. TIM-3 binds to a wide range of ligands that are often up-regulated in tumor microenvironments (Table 1).

TIM-3 is constitutively expressed on most CD56^dim^CD16^+^ NK cells and a fraction of CD56^bright^CD16^-^ NK cells in the peripheral blood of healthy individuals [161]. Furthermore, cytokine stimulation has been shown to increase levels of TIM-3 expression on both CD56^dim^ and CD56^bright^ NK cell subsets [161,162]. When compared to healthy donors, increased TIM-3 expression has been observed on peripheral blood NK cells from patients with lung adenocarcinoma [163], melanoma [164], gastric [165], and bladder cancers [166]. In melanoma and lung cancer, TIM-3 expression on NK cells increases with disease stage [163,164,167], and higher percentages of TIM-3^+^ NK cells have been associated with lower survival of lung adenocarcinoma patients [163].

Despite being a marker of exhausted T cells [168], TIM-3 expression is not associated with dysfunction of healthy donor NK cells. On the contrary, TIM-3 is expressed on functional human NK cells that degranulate and kill K562 cells and produce IFN-γ upon in vitro cytokine stimulation [161,169,170]. The cytolytic activity of TIM-3^+^ NK cells from healthy donors appears even greater than their TIM-3^-^ counterparts [161,170]. Although a study investigating esophageal cancer reported decreased IFN-γ production and degranulation capacity of tumor-infiltrating TIM-3^+^ NK cells compared to their TIM-3^-^ counterparts, it should be noted that these experiments were performed using unphysiological stimulation with PMA and ionomycin [171].

Initial work by Gleason et al. suggested that, in human NK cells, TIM-3 might act as an activating receptor and promote NK cell production of IFN-γ upon binding to its ligand Galectin 9 [162]. This was demonstrated using the NK92 NK cell line engineered to overexpress TIM-3 as well as primary TIM-3^+^ NK cells that were found to specifically produce IFN-γ in response to recombinant human Galectin 9 in vitro. These findings were soon challenged by Ndhlovu et al., who used redirected killing assay against P815 cells to show that TIM-3 cross-linking suppresses the cytotoxicity of healthy donor NK cells [161]. These discrepancies might be explained by the hypothesis that TIM-3 could mediate both positive and negative signaling depending on the ligand recognized. Still, most reports suggest that TIM-3 inhibits NK cell functions in cancer patients. For instance, in vitro blockade with anti-TIM-3 mAbs improves the cytotoxicity, production of IFN-γ, and proliferation of NK cells from melanoma patients [164] and the cytotoxicity of lung cancer patient NK cells [163]. In conclusion, TIM-3 is expressed on fully functional mature and/or activated NK cells and might function as an inhibitory receptor that restrains NK cell function similarly to KIRs or NKG2A.

## 6. Other Inhibitory Pathways

### 6.1. Adenosinergic Signaling

Adenosine is an immunosuppressive metabolite generated from extracellular ATP by the subsequent enzymatic activities of CD39 and CD73 [172]. Mouse and human NK cells express the high affinity adenosine 2A receptor (A2AR) [173]. Adenosine signaling through A2AR was shown to limit the expansion of mouse NK cells, in vitro and in vivo [173]. The use of Adora2a^floxflox^;Ncr1^iCre^ mice (in which A2AR is selectively deleted in NK cells) established that A2AR directly inhibits NK cell anti-tumor activity against transplantable SM1WT1 BRAF-mutant melanoma [173]. A2AR signaling was also found to limit mouse NK cell ADCC activity in vitro and in vivo [174]. Surprisingly, Chambers et al. reported that the addition of adenosine to human NK cells cultured with IL-12 and IL-15 resulted in increased IFN-γ production while adenosine down-regulated NK cell metabolism [175]. These results suggest that extracellular adenosine acts on specific cellular pathways rather than causing general NK cell inhibition. Overall, CD39, CD73, and A2AR appear as promising targets for inhibitory drugs aiming at enhancing NK cell activity in cancer.

### 6.2. Interleukin-1 Receptor 8 (IL-1R8)

IL-1R8 (SIGIRR or TIR8) is a member of the IL-1 receptor family expressed at high levels on mouse and human NK cells [140]. Genetic ablation of Il1r8 in mice results in increased NK cell maturation as well as increased NK cell production of IFN-γ in response to in vitro stimulation with IL-12 and IL-18 [140]. Importantly, compared to WT mice, Ilr8^−/−^ mice are resistant to liver carcinogenesis and metastasis, a protection that was shown to be NK cell-dependent [140]. Overall, data to date point to IL1R8 as a new negative regulator of IL-18 signaling in NK cells that could be targeted to unleash NK cell activity, with expected enhanced benefits if IL-18 is expressed within the tumor microenvironment [176].

### 6.3. KIR3DL3

Most inhibitory KIRs recognize MHC-I molecules but KIR3DL3 is an exception, the ligand of which has remained elusive until recently [177]. Two recent studies indicated that KIR3DL3 might down-regulate NK cell functions through binding to HHLA2 (also called B7-H7), a B7 family molecule expressed on antigen-presenting cells and highly expressed in many human cancers [178,179]. HLLA2 is known to bind a transmembrane and immunoglobulin domain containing 2 (TMIGD2, also called IGPR-1 or CD28H), a co-stimulatory receptor expressed on NK cells and T cells. Interestingly, TMIGD2 is expressed on 50–90% peripheral blood NK cells, with higher expression in the CD56^bright^ CD16^-^ subset [178]. By contrast, KIR3DL3 is preferentially expressed on CD56^dim^ CD16^+^ NK cells [178]. Bhatt et al. used the NK92 MI cell line to show that overexpression of HLLA2 on tumor cells inhibits in vitro NK cell-mediated killing through HLLA2-KIR3DL2 interactions [179]. Cross-linking of KIRDL3 was also shown to increase the redirected lysis of P815 cells and the cytokine production of healthy donor NK cells [178]. Using immunodeficient mice challenged with the HHLA2^+^ human lung cancer cell HCC827 and injected with KIR3DL3^+^-expanded human NK cells, Wei et al. demonstrated that KIR3DL3 blockade enhances NK cell-mediated protection in vivo. While these new studies picture KIR3DL3 as a promising therapeutic target, further work should investigate how KIR3DL3 and TMIGD2 co-regulate NK cell responses to HLLA2^+^ tumors.

### 6.4. CD161

CD161 (also called NKR-P1A or KLRB1) is a C-type, lectin-like glycoprotein expressed on human NK cells and T cells [138]. CD161 binds to Lectin-like transcript 1 (LLT1, also called CLEC2D or OCIL) that is broadly expressed on cells of the hematopoietic lineage, including B cell-derived cancer cell lines [180,181]. Contrasting with CD161 on T cells, which has been shown to enhance cytokine production following TCR triggering, CD161 on NK cells appears to act as an inhibitory receptor [182]. Transfection of LLT1 into NK cell targets protects them from human NK cell cytotoxicity in a CD161-dependent manner [180,183]. In vitro studies also suggest that LLT1 expression in human malignant glioblastoma cells contributes to tumor escape by inhibiting NK cell-mediated lysis [184]. Similar data have been obtained with human breast cancer cell lines [185]. Rahim et al. established a role for the mouse NKR-NKRP1B:Clr-b axis (murine equivalent to human CD161:LLT1) in mediating NK cell tolerance, as Nkrp1b^−/−^ mouse NK cells show increased lysis of LLT1-expressing target cells when compared to WT NK cells [186]. This group also observed delayed onset of transplantable Eµ-Myc B cell lymphoma in Nkrp1b^−/−^ mice, indicating that NKRP1B might limit tumor growth in vivo [186]. Similarly, Tanaka et al. used Eµ-Myc transgenic mice (spontaneously developing B cell lymphoma) backcrossed into a Clr-b^−/−^ background to confirm that the NKR-P1B:Clr-b recognition axis may impair NK cell-mediated immunosurveillance [187]. Based on these results, the authors suggested that the CD161:LLT1 system may represent a target for immune checkpoint therapy. However, caution should be exerted when translating these findings to humans since opposing data have been reported in cancer patients. Indeed, CLEC2D and KLRB1 gene expressions (encoding for LLT1 and CD161, respectively) have been associated with better clinical outcomes for lung cancer patients, an observation that may be explained by the increased IFN-γ production of tumor-infiltrating CD161^+^ CD4^+^ T cells when compared to their CD161^-^ counterparts [188]. Further work should determine whether these discrepancies stem from interspecies differences or whether they reflect different immune responses to diverse cancer types (e.g., LLT1-expressing lymphoma controlled by NK cells and solid tumor mostly controlled by T cells). Moreover, LLT1 is expressed on human NK cell lines and its cross-linking has been shown to stimulate IFN-γ production [189,190]. This dual role of the LLT1-CD161 axis on NK cells, together with the apparent opposite functions of CD161 on NK cells and T cells, would need to be carefully considered if this pathway was to be targeted in cancer patients.

### 6.5. Siglec Receptors

Siglecs comprise a lectin family of surface receptors that bind to sialic acid-containing carbohydrates (sialoglycans) and mediate activating or inhibitory signals [191]. Sialic acids are expressed on healthy cells but hypersialylation is considered a hallmark of cancer, and engagement of Siglec receptors by tumor sialoglycans regulates immune recognition of cancer cells [191]. Human NK cells express two inhibitory Siglec receptors that signal through ITIM motifs: Siglec-7 and Siglec-9 [191]. Siglec-7 (CD328, p75/AIRM1) is expressed on the majority of healthy donor peripheral blood NK cells, while Siglec-9 expression is restricted to a subset of CD56^dim^ NK cells [192,193]. Interestingly, reduced percentages of Siglec-9^+^ NK cells have been reported in the blood of malignant melanoma or colon adenocarcinoma patients when compared with healthy donors [193]. Moreover, when compared with their Siglec-9-negative counterparts, healthy donor CD56^dim^ Siglec-9^+^ NK cells exhibit decreased killing of K562 target cells [193]. 

Ligands for Siglec-7 and -9 are widely expressed on human tumor cells of different histological origins [193]. Most specifically, in vitro studies suggested that binding of Siglec-7 to the gangliosides DSGb5 and GD3 (often overexpressed in cancers such as renal cell carcinoma or melanoma) might reduce NK cell cytotoxicity [151,152]. However, the importance of this mechanism for cancer immunosurveillance is unclear since sialidase treatment is required to unmask the sialic acid-binding site of Siglec-7. Further work is needed to establish the natural conditions unmasking Siglec-7 on NK cells that might allow Siglec-7 binding to tumor cell-expressed ligands in vivo. The cell surface mucin MUC16, expressed at high levels by epithelial ovarian tumors, has been identified as a ligand for Siglec-9; however, its ability to modulate NK cell response remains to be evaluated [154]. More recently, Jandus et al. elegantly used neuraminidase to assess the broad effect of Siglec ligands on NK cell activity [193]. Neuraminidases strip sialic acids from the cell surface, thereby removing potential Siglec ligands. The authors showed that neuraminidase treatment of target cells expressing ligands of Siglec-7 and -9 (HeLa and K562 cell lines) enhances their killing by human NK cells. This inhibitory effect or Siglec ligands on NK cell activity was demonstrated in vitro, using standard cytotoxicity assays but also in vivo, using humanized mice (e.g., mice with a reconstituted human NK cell compartment) [193]. High-affinity molecules that can disrupt the binding of Siglec-7 to its ligands have been proposed as a new therapeutic approach to counter cancer immune evasion [194,195]. Yet, more insights on Siglec-mediated modulation of NK cell functions in vivo would be needed to evaluate the benefits of targeting this pathway in cancer patients. Given the absence of mouse homologues for Siglec-7/9, humanized mouse models, as developed by Jandus et al. [193], represent essential tools for the field. 

### 6.6. CD300a

CD300a (also called Inhibitory receptor protein 60, Irp60) and CD300c are paired receptors recognizing phosphatidylethanolamine and phosphatidylserine, two phospholipids normally expressed on the cytosolic surface of the plasma membrane and externalized on apoptotic cells or on some tumor cells [143] of the activating receptor. CD300c is restricted to IL-2- or IL-15-activated CD56^bright^ NK cells [143]. By contrast, CD300a is expressed by the majority of CD56^dim^ and CD56^bright^ NK cells [196]. CD300a exhibits four ITIMs domains that mediate inhibitory signaling [197]. In vitro experiments assessing direct cytotoxicity as well as redirected lysis against P815 target cells demonstrated that CD300a cross-linking decreases NK cell killing [198,199]. Importantly, blocking the interaction of CD300a with phosphatidylserine was shown to enhance the killing of RKO or 293T tumor cells by bulk human NK cell cultures [199]. CD300a has been suggested as a target to increase NK cell functions [28,200]. Still, further work will be required to evaluate the importance of CD300a in regulating NK cell recognition of tumor cells in vivo.

## 7. Conclusions

Due to the success of immune checkpoint blockade and the complementarity of NK cells and T cells in the fight against cancer, NK cell immune checkpoints have recently attracted a lot of attention. Immune checkpoints are commonly up-regulated on exhausted T cells. NK cells and T cells might share an epigenetic program of exhaustion, which is associated with the expression of the immune checkpoints PD-1 and LAG-3 [129,201]. However, TIM-3 and TIGIT, two other common T cell exhaustion markers, are expressed at high levels on unstimulated and functional human NK cells [50,162]. A better understanding of how these inhibitory pathways might be differentially regulated in NK cells and T cells would be beneficial to design immunotherapies invigorating both innate and adaptive cytotoxic lymphocytes in cancer patients (Figure 2).

## Figures and Tables

**Figure 1 cancers-13-04263-f001:**
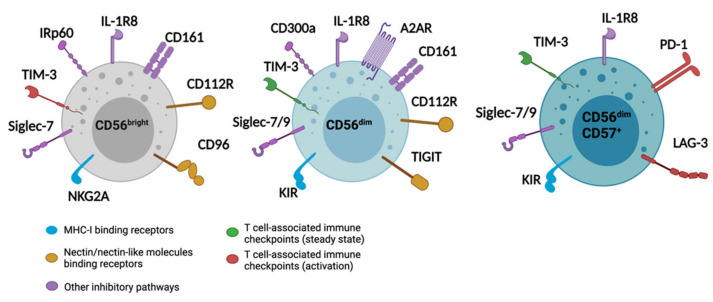
Differential expression of inhibitory receptors NK cell subsets. Immature CD56^bright^, mature CD56^dim^, and terminally differentiated CD56^dim^CD57^+^ NK cells are regulated by different inhibitory receptors. MHC-I binding receptors are represented in blue, receptors binding to nectin/nectin-like molecules binding are in yellow, T cell-associated immune checkpoints expressed in steady state are in green, T cell-associated immune checkpoints up-regulated upon activation are in red, and other inhibitory pathways are shown in purple.

**Figure 2 cancers-13-04263-f002:**
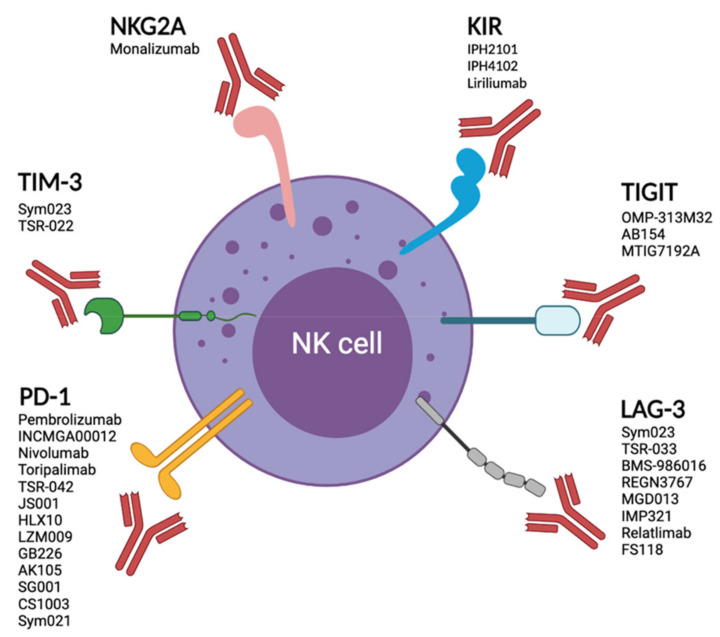
Enhancing NK cell functions through immune checkpoint blockade. Several immunotherapeutic approaches are currently being explored to improve NK cell effector function. Immune checkpoint blockade may result in increased NK cell numbers (in the case of anti-PD-1 therapy), increased NK cell cytotoxicity, and/or cytokine production such as IFN-γ. A vast majority of immune checkpoint therapies are delivered in combination with other agents. For example, anti-NKG2A mAb (monalizumab) has been tested as a single agent for CLL patients (NCT02557516) and in combination with anti-PD-L1 (Durvalumab) for metastatic colorectal cancer (NCT02671435). KIR blockade monotherapy was used in multiple myeloma (NCT00552396, NCT00999830), or in combination with anti-PD-L1 (nivolumab) for myelodysplastic syndrome (NCT02599649), also in combination with anti PD-1 Ab (Nivolumab) and anti-CTLA-4 mAb (Ipilimumab) to treat advanced solid tumors (NCT03203876). Other examples include anti-TIGIT (MTIG7192A) together with anti-PD-1 (Atezolizumab) to treat non-small cell lung cancer (NCT03563716), or anti-LAG3 (IMP321) and anti-PD-1 (Pembrolizumab) for treatment of melanoma patients (NCT02676869). Several TIM-3 blockade clinical trials are currently underway (NCT03311412, NCT03066648, NCT03489343, NCT03680508, NCT03961971, NCT04370704, NCT02817633, NCT03099109, NCT03744468, and NCT04139902). In addition, Lag-3 blockade is currently being tested in a range of tumors (NCT01968109, NCT04150965, NCT04140500, and NCT03625323).

**Table 1 cancers-13-04263-t001:** Checkpoint receptors expressing cells and corresponding ligands.

Receptor	Expression	Ligands	Refs
A2AR	T, NK cells	Adenosine	[136]
Unknown receptor to B7-H3		B7-H3	[124]
BTLA	T, B, NK cells, DCs and myeloid cells	HVEM	[112,137]
CD112R	T, NK cells	CD112	[72]
CD161	T, NK cells	Lectin-like transcript 1 (LLT1, also called CLEC2D or OCIL)	[138]
CD96	T, NK, NKT cells	Human: CD155 (PVR, Necl-5) Mouse: CD155 and Nectin-1	[68,139]
CTLA-4	Regulatory T cells (Tregs), activated T and B cells, activated murine NK cells, human NK cells?	CD80 (B7-1), CD86 (B7-2)	[96]
IL-1R8	Monocytes, B, T, NK cells, DCs	IL-37	[140,141,142]
Irp60	Mast cells, NK cells	Phosphatidylethanolamine (PE) and phosphatidylserine (PtdSer)	[143,144]
KIRs	NK cells, CD8+ T cells	HLA-ABC	[145]
LAG-3	Regulatory T cells (Tregs), activated T and NK cells, B cells, NKT cells, pDCs	MHC II; Galectin-3 (Gal-3); LSECtin fibrinogen-like protein 1 (FGL1);	[146,147,148,149]
NKG2A	NK cells, CD8+ T cells	HLA-E	[150]
PD-1	Activated T and B cells, NK cells, NKT cells, ILC-2 cells, myeloid cells	PD-L1 (B7-H1); PD-L2 (B7-DC)	[78]
Siglec-7	Monocytes, CD8+ T and NK cells	DSGb5, GD3	[151,152,153]
Siglec-9	T, B, NK cells (CD56dim), granulocytes and monocytes	MUC16	[154,155]
TIGIT	T, NK, NKT cells	CD155, CD112, CD113	[58,61]
TIM-3	T, NK, NKT cells and myeloid cells	Galectin-9 (Gal-9) Phosphatidylserine (PtdSer); High mobility group protein B1- HMGB1; Carcinoembryonic antigen cell adhesion molecule 1 CEACAM-1	[156,157]
VISTA	Myeloid cells, TCRγδ T cells, naïve CD4+ and CD8+ TCRαβ T cells, Tregs	PSGL-1, VSIG3	[158,159,160]

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
