# Peer review of "Inhibitory Receptors and Immune Checkpoints Regulating Natural Killer Cell Responses to Cancer"

_cancers, 2021, doi:10.3390/cancers13174263_

Round 1

Reviewer 1 Report

Comments:

This is timely review and will be of interest to the readers.

In this article authors reviewed regulation of immune checkpoints mainly associated with natural killer (NK) cells and T-cells for cancer immunotherapy.  

Authors chose to review very important topic attributes to the development of new therapeutic targets for cancer treatment.  

Authors discussed regulation of NK cell functions in cancer through inhibitory receptors including immune checkpoints.

Authors discussed some of elegant mouse genetic studies in the current review. Below are the some of the important studies discussed in this review:

  1. Nkg7-/- NK cells were shown to exhibit impaired degranulation against tumor cells, resulting in decreased in vivo protection against experimental B16F10 or LWT1 metastasis.
  2. The use of Adora2a floxflox; Ncr1iCre mice (in which A2AR is selectively deleted in NK cells) established that A2AR directly inhibits NK cell anti-tumor activity against transplantable SM1WT1 BRAF-mutant melanoma.
  3. Genetic ablation of Il1r8 in mice results in increased NK cell maturation as well as increased NK cell production of IFN-γ in response to in vitro stimulation with IL-12 and IL-18.

Authors elaborated on inhibitory receptors that recognize the molecules of the nectin and nectin-like family such as TIGIT or CD96, as well as immune checkpoints described on T cells including PD-1 and TIM-3.

Authors also discussed recently discovered pathways and receptors including Siglecs and Irp60.

Moreover, authors clearly state the future studies required to bridge the gap in the literature.

Together, all these aspects add strength when compared with the other published articles.

Conclusions supported the main question addressed in this review article.

Minor points:

Check for extra use of hyphen. Example: accumu-lating, discov-ered etc.

Author Response

We thank the reviewer for their positive comments.

We have checked the extra hyphens and they seem to be automatically generated by the template (when a word is split between 2 lines). 

Reviewer 2 Report

The authors explore the role of inhibitory receptors in NK cells in the context of cancer, describing the characterization, functions, and targeting approaches of well-described as well as emerging NK cell inhibitory signals.

The review is detailed, comprehensive and well written.

Minor points:

  • I would suggest editing the structure in the first part (Paragraphs 2 and 3, and Figure 1). The description of the recent discovery about NKG7 doesn’t really fit in the paragraph about “basis of NK cell activation”. I would move it if possible. I would put Figure 1 later in the text, after the receptors have been mentioned. I would change the title of paragraph 3, which is mostly about MHCI binding receptors.
  • Please check line 86 (page 2): “..introduce the role of..”
  • I would suggest mentioning the recently described role of KIR3DL3 in NK cells (Bhatt et al, 2021; Wei et al, 2021).

Author Response

We thank the reviewer for their positive comment and suggestions.

Regarding the NKG7 paper, we (and reviewer 1) think that this paper needs to be cited; but it did not fit better in another paragraph. Therefore, we decided to keep NKG7 discussion in paragraph 2 but changed the paragraph title to:  “Regulation of NK cell activation and cytotoxic activity”.

Figure 1 has been moved according to the reviewer’s suggestion.

We have changed the title of paragraph 3 according to the reviewer’s suggestion (thanks for noticing!).

Thanks for noticing the typo page 2, we have corrected it.

Thanks for the KIR3DL3 suggestion. This has now been included into the manuscript (new section 6.3) and the suggested references have been cited.